# Vaccines in Dermatology—Present and Future: A Review

**DOI:** 10.3390/vaccines13020125

**Published:** 2025-01-26

**Authors:** Eyan Goh, Jean-Marc Chavatte, Raymond T. P. Lin, Lisa F. P. Ng, Laurent Rénia, Hazel H. Oon

**Affiliations:** 1Lee Kong Chian School of Medicine, Nanyang Technological University, Singapore 308232, Singapore; egoh031@e.ntu.edu.sg (E.G.); lisa_ng@idlabs.a-star.edu.sg (L.F.P.N.); renia_laurent@idlabs.a-star.edu.sg (L.R.); 2National Public Health Laboratory, Singapore 308442, Singapore; jean-marc_chavatte@moh.gov.sg (J.-M.C.); micltprv@nus.edu.sg (R.T.P.L.); 3National University Hospital Singapore, Singapore 119077, Singapore; 4A*STAR Infectious Diseases Labs (A*STAR IDL), Agency for Science, Technology, and Research (A*STAR), Singapore 138648, Singapore; 5Yong Loo Lin School of Medicine, National University of Singapore, Singapore 117597, Singapore; 6School of Biological Sciences, Nanyang Technological University, Singapore 637551, Singapore; 7National Skin Centre and Skin Research Institute of Singapore, Singapore 308205, Singapore

**Keywords:** vaccines, immunization, dermatology

## Abstract

Dermatological vaccines have emerged as critical tools in preventing and managing a wide spectrum of skin conditions ranging from infectious diseases to malignancies. By synthesizing evidence from existing literature, this review aims to comprehensively evaluate the efficacy, safety, and immunogenicity of vaccines used in dermatology, including both approved vaccines and those currently being researched. Vaccines discussed in this paper include those targeting dermatoses and malignancies (e.g., acne vulgaris, atopic dermatitis, and melanoma); infectious diseases (e.g., human papillomavirus (HPV); varicella zoster virus (VZV); herpes zoster (HZ); warts; smallpox; mpox (monkeypox); hand, foot, and mouth disease (HFMD); candidiasis and Group B Streptococcus (GBS); and neglected tropical diseases (e.g., Buruli ulcer, leprosy, and leishmaniasis). Through this review, we aim to provide a detailed understanding of the role of vaccines in dermatology, identify knowledge gaps, and propose areas for future research.

## 1. Introduction

Vaccines play a crucial role in preventing a wide range of diseases and have become a cornerstone of medicine and public health. Today, the World Health Organization (WHO) recognizes over 40 human diseases for which vaccines are either available or are currently in development [1]. This article will review vaccines of dermatologic interest and provide an overview of their clinical efficacy and summarize ongoing research efforts.

## 2. Materials and Methods

### 2.1. Search Strategy

A literature search was conducted in the electronic databases PubMed, clinicaltrials.gov, and Embase to identify studies on vaccines related to dermatological conditions. The search was limited to articles published between January 2013 and June 2024 to ensure relevance. Search terms included “vaccines”, “vaccination”, “immunization”, “skin diseases”, “dermatology”, “acne vulgaris”, “Propionibacterium”, “*Propionibacterium acnes*”, “Cutibacterium”, “*Cutibacterium acnes*”, “human papillomavirus viruses”, “varicella zoster virus infection”, “herpes zoster”, “melanoma”, “dermatitis”, “warts”, “candidiasis, chronic mucocutaneous”, “neglected diseases”, “Buruli ulcer”, “leprosy”, “leishmaniasis”, “mpox (monkeypox)”, “hand, foot, and mouth disease”, and “*Streptococcus agalactiae*”, combined with Boolean operators AND, OR, and NOT as appropriate. All articles were screened based on the content of the title and abstract to select the studies that assessed the efficacy and safety of dermatological vaccines.

### 2.2. Inclusion Criteria

Studies were included if they met the following criteria:Conditions targeted: vaccines designed to prevent or treat dermatological conditions.Population: studies investigating the efficacy and safety of dermatological vaccines in human populations without known underlying health conditions, as well as animal populations specifically for acne vulgaris research.Study types: clinical trials and randomized controlled trials (RCTs). For acne vulgaris specifically, the inclusion criteria were expanded to include review articles due to the absence of clinical trials or RCTs found in the database search.Publication date: studies published within the last 10 years (January 2013 to June 2024).Language: studies published in English to ensure accessibility.

### 2.3. Exclusion Criteria

Studies were excluded based on the following criteria:Irrelevant conditions: vaccines not targeting dermatological conditions;Population: studies focusing on the efficacy of vaccines in special populations, such as those involving stem cell transplant recipients, individuals with diabetes, or those who are immunodeficient (to ensure generalizability of findings to the broader population without underlying health conditions);Type of vaccine: studies evaluating dermatological vaccines that were co-administered with other vaccines were excluded to avoid confounding effects on vaccine efficacy and safety outcomes.

## 3. Vaccines

### 3.1. Acne vulgaris

Acne vulgaris is one of the most prevalent diseases worldwide, affecting up to 90% of adolescents. It accounts for 4.96 million disability-adjusted life years (DALYs) and is the second largest contributor of combined skin age-standardized DALYs [2]. However, in spite of the plethora of acne consumer products available, these products are developed primarily for the treatment of active acne and acne scarring, rather than for disease prevention. Previous research has established that an interplay between hormonal influences on sebum production, the presence of *Cutibacterium acnes* (*C. acnes*) bacteria in the pilosebaceous follicle microbiome, and dysregulated cellular immune responses underlies the pathogenesis of acne vulgaris [3]. Although the exact role *C. acnes* plays in the pathogenesis of acne remains disputed, existing literature suggests that *C. acnes* contributes to acne development by inducing both innate and adaptive immune responses [4].

#### 3.1.1. The Challenges of Assessing Acne Vaccines

Currently, the assessment of acne treatment efficacy is complicated by the variety of outcome measures [5]. The lack of a standardized set of outcome measures makes it challenging to ensure clinical evidence is reliable. Historically, physician assessments like lesion counts and standardized acne severity scales like the Global Acne Grading System (GAGS) and the Investigator’s Global Assessment (IGA) have been used, including in the two ongoing clinical trials mentioned below [6,7]. However, there is growing recognition of the importance of considering the broader impact of acne, as well as the specific treatment modality, on patients. In response, the Acne Core Outcomes Research Network (ACORN) has developed a set of core measures, including clinical improvements and seven key patient-reported outcome domains: satisfaction with appearance, extent of scars or dark marks, long-term control, signs and symptoms, satisfaction with treatment, health-related quality of life, and adverse events [8]. These parameters will be essential for evaluating the efficacy of new acne treatments as they progress through clinical trials, leading to the development of more directed and effective anti-acne therapies.

#### 3.1.2. Vaccine Platforms

After the publication of the complete genome sequence of *C. acnes* in 2004, subsequent research has linked several putative virulence factors to host inflammation and acne development, marking these as potential targets for vaccine development [9]. These virulence factors, including sialidases, neuraminidases, endoglycoceramidases, adhesins, thermal shock proteins, Christie–Atkins–Munch–Petersen (CAMP) factors, lipases, and esterases, are involved in diverse mechanisms such as cell adhesion, inflammation, tissue invasion and degradation, and the synthesis of capsule polysaccharides [10].

In a pre-clinical trial in a mouse model (Nakatsuji et al., 2008), intranasal administration of inactivated *C. acnes*—previously demonstrated to induce IL-8 production by sebocytes—led to ear inflammation in the immunized mice [11]. This approach could be promising for acne treatment, as it suggests that targeted immunization might modulate the immune response against *C. acnes*, potentially reducing its pathogenic impact and associated inflammation. However, it is important to note that *C. acnes* is a commensal bacterium that plays a crucial role in maintaining skin health by inhibiting the invasion of common pathogens such as *Staphylococcus aureus* and *Streptococcus pyogenes* [12]. Disrupting this balance could undermine the skin’s natural defense mechanisms, potentially doing more harm than good. As such, more research is required to clearly delineate the impact of such vaccines on overall skin health.

Subsequently, Nakatsuji et al. (2008) designed an acne vaccine targeting a cell wall-anchored sialidase [13]. Sialidases present in *C. acnes* are thought to be involved in the adhesion of *C. acnes* to sebocytes and the cleavage of sialoglycoconjugates, producing sialic acids that are then utilized as substrates for energy generation [14]. Hence, inhibition of sialidases may prevent adhesion of *C. acnes* to host tissues and limit the availability of sialic acids as nutrients. The study found that the vaccine effectively induced antibody production in mice, which suppressed *C. acnes*-induced inflammation both in vitro and in vivo. These antibodies attenuated *C. acnes*-induced cytotoxicity and IL-8 production in human sebocytes in vitro. Moreover, immunization reduced ear swelling and inhibited the release of pro-inflammatory macrophage inflammatory protein (MIP-2) cytokine in immunized mice compared to controls after challenge with *C. acnes* [13].

Evidence suggests that targeting virulence factors secreted by *C. acnes* could minimize disruption of the microbiome [15] and exert less selective pressure, potentially reducing the risk of microbial resistance [16]. As such, Wang et al. (2018) generated a vaccine targeting CAMP factor, a toxin that forms pores in host membranes and leads to inflammation and host tissue damage. The vaccine was found to reduce *C. acnes* colonization and inflammation; vaccine-induced antibodies against CAMP factor suppressed the inflammatory response to *C. acnes* in mice and ex vivo in human acne explants, suggesting a role for CAMP factor in inflammation. The reduction in inflammation correlated with reduced levels of chemokines such as MIP-2 in mice and lower levels of IL-8 and IL-1β in a human explant model [17]. Similarly, Liu et al. (2011) found that CAMP2-neutralising antibodies significantly decreased the inflammation induced by *C. acnes* in a mouse ear model [15]. More recently, a peptide vaccine targeting HylA, a virulence factor expressed by *C. acnes* strains associated with acne, successfully attenuated HylA-induced inflammation in mouse models [18]. In view of these findings, vaccines targeting *C. acnes* CAMP factor and HylA could be promising.

Ongoing clinical trials are being performed in humans. The first sought to evaluate the safety, tolerability, and immunogenicity of an acne vulgaris vaccine (ORI-A-ce001) based on recombinant proteins (which have not been publicly disclosed) in 38 healthy subjects aged 18 and above with moderate facial acne vulgaris [6]. The second study is a phase 1/2 trial focusing on delineating the safety, efficacy, and immunogenicity of an acne mRNA vaccine candidate at three different dose levels in adults aged 18 to 45 with moderate to severe acne. The study began in April 2024 and is expected to be completed by December 2027 [7]. A phase 1 trial in mild acne is anticipated to start in Singapore in Q2 2025 with a translational study.

Targeting virulence factors through vaccination could potentially circumvent the drawbacks associated with antibiotics and other acne treatment options, such as the lack of specificity, emergence of antibiotic-resistant bacterial strains, and other adverse side effects. Current vaccine platforms for acne vulgaris are summarized in Table 1. As more virulence factors are uncovered, new vaccine targets may emerge. However, it remains uncertain whether the results from animal models will translate effectively to humans; further clinical trials are warranted to evaluate the efficacy of these vaccines in humans. Additionally, the potential for cross-reactivity of these antibodies with other proteins or cells requires careful investigation. Finally, while these vaccines are currently designed for therapeutic purposes in individuals with acne to reduce disease severity, the possibility of using them as a preventive intervention remains to be explored. Nonetheless, ongoing research efforts offer hope for the future availability of preventative and therapeutic interventions for acne.

### 3.2. HPV

Human papillomavirus (HPV) is the most common sexually transmitted infection (STI), representing a significant global public health concern due to its links to various skin and mucous membrane conditions, such as genital warts and cervical cancer [19]. Of more than 200 genotypes identified, over 40 are capable of infecting the genital areas of both men and women, including the skin of the penis, vulva, and anus, and the linings of the vagina, cervix, and rectum [20]. Anogenital warts, or condylomata acuminata, are frequently caused by strains of HPV-6 and -11; on the other hand, subtypes HPV-16, -18, -31, -33, -35, -39, -45, -51, -52, -56, -58, -59, -66, and -68 are associated with the development of malignant lesions, including invasive cervical carcinomas in women, as well as cancers of the vulva, vagina, and anus [19,20]. Notably, HPV-6 and HPV-11 account for over 90% of genital warts cases [21], while the most carcinogenic strains, HPV-16 and HPV-18, predominate among those causing cancer [22]. In addition, non-genital warts caused by HPV infection are exceedingly common and are a significant source of morbidity [23].

#### 3.2.1. Vaccine Platforms

Currently, there are three vaccines available: a bivalent vaccine targeting HPV types 16 and 18; a quadrivalent vaccine targeting HPV types 6, 11, 16, and 18; and a newer nonavalent vaccine targeting HPV types 6, 11, 16, 18, 31, 33, 45, 52, and 58. These vaccines were developed using recombinant technology, utilizing the HPV L1 protein to form virus-like particles (VLPs) that mimic the natural virus structure, which induce the production of antibodies targeting type-specific L1 proteins. Prepared from purified L1 structural proteins, none of these vaccines contain live biological products or viral DNA, ensuring they are non-infectious. These vaccines are formulated with an adjuvant (AS04) to enhance the immune response. They are administered via intramuscular (IM) injection on a three-dose schedule: at 0, 1, and 6 months for the bivalent vaccine, and at 0, 2, and 6 months for the quadrivalent vaccine [24].

Since they offer a high level of protection against the most common high-risk HPV strains associated with cervical cancer (HPV types 16 and 18) and low-risk strains linked to genital warts (HPV types 6 and 11), HPV vaccines have been integrated into the national immunization programs of over 100 countries as of 2020 [25]. The HPV vaccines are among the safest and most effective vaccines available, with numerous large-scale studies demonstrating their safety and efficacy in preventing HPV infection and related diseases.

Two extensive randomized trials, encompassing over 17,000 young women, scrutinized the efficacy of the quadrivalent vaccine in treating anogenital lesions. In a trial by the FUTURE II Study Group, the vaccine demonstrated 98% efficacy for the prevention of cervical intraepithelial neoplasia (CIN) grade 2 or 3, adenocarcinoma in situ, or invasive carcinoma of the cervix caused by HPV-16/-18 among women without previous exposure to these HPV types. However, in the intention-to-treat analysis, the efficacy was lower at 44%, primarily attributed to the fact that this analysis included participants who had previously been exposed to HPV-16/-18 prior to the first injection [26]. In the FUTURE I trial, the vaccine demonstrated 100% efficacy in preventing external anogenital and vaginal lesions caused by the HPV strains covered by the vaccine among women with no prior HPV exposure. However, in the intention-to-treat analysis, the efficacy stood at 73% for the HPV types targeted by the vaccine and 34% for lesions associated with any HPV type [27]. These findings align with previous results showing that intralesional administration of the quadrivalent HPV vaccine effectively treated genital warts with a 91.7% success rate [28]. Long-term follow-up data have further substantiated the efficacy and safety of the quadrivalent vaccine in adult women. In an extended-follow-up phase, there were no reported cases of HPV-6/-11/-16/-18-related CIN or external genital lesions in the per-protocol population. In addition, immunogenicity against the vaccine-related HPV types persisted, with no evidence of HPV type replacement. Furthermore, no new serious adverse experiences were reported [29].

The bivalent vaccine also exhibited remarkable efficacy, ranging from 90% to 100%, in preventing incident infections by HPV types 16 and 18 [30,31], as well as CIN lesions [32], while maintaining a favorable safety profile. Notably, in the PATRICIA trial, the vaccine demonstrated 100% efficacy for the prevention of CIN3+ associated with HPV-16/-18 among women with no evidence of oncogenic HPV infection at baseline and 45.7% in the total vaccinated cohort [33].

Studies evaluating the efficacy of the HPV vaccines in treating non-genital warts have yielded variable results. In one study on recalcitrant non-genital warts, of the 19 patients who received three doses of the quadrivalent vaccine, 26% achieved complete clearance, 42% achieved partial clearance, and 32% showed no response. Among the four patients who received two doses, 75% achieved complete clearance, while 25% achieved partial clearance. None of the three patients who received only one dose demonstrated significant improvement, and no adverse reactions related to the vaccine were reported [34]. Furthermore, Hayashi et al. found that administration of the quadrivalent vaccine resulted in complete resolution of multiple warts in two out of three patients despite HPV genotyping revealing subtypes not directly targeted by the vaccine, suggesting the possibility of cross-protective immunity [35].

In a more recent study evaluating both the quadrivalent and bivalent vaccines, the quadrivalent vaccine demonstrated a notable efficacy, with complete clearance of warts observed in 18 out of 20 patients (90%); in contrast, only 6 out of 20 patients (30%) receiving the bivalent vaccine achieved complete clearance, while no response was reported in the saline control group [36]. However, a previous study found that the intralesional bivalent HPV vaccine treated recalcitrant common warts with a much higher success rate of 81.8% [37]. The discrepancy in clearance rates between this and prior studies may be due to variations in the injected volume (0.3 mL vs. 0.1 mL) and the types of warts treated.

Taken in totality, these findings suggest that both the bivalent and quadrivalent vaccines may have a promising role in the treatment of recalcitrant extragenital warts.

The nonavalent vaccine was evaluated in a study comparing single-dose and two-dose regimens for HPV vaccination in the UK. Implementation of the single-dose nonavalent HPV vaccination, in line with the UK’s single-dose policy (effective September 2023), was analyzed by an age-structured dynamic model of HPV transmission. This approach was found to be cost-effective compared to no vaccination, with an incremental cost-effectiveness ratio below GBP 2040/QALY [38].

#### 3.2.2. Indications and Contraindications

Given that HPV is implicated in virtually all invasive cervical cancers, as well as in precancerous lesions and genital warts, the primary objective of HPV vaccination is to prevent genital infections and the associated morbidity [39].

HPV vaccines are primarily indicated for individuals aged 9 to 45 years and are most effective when administered before the onset of sexual activity, ideally before first exposure to HPV infection. However, sexually active individuals can still benefit from vaccination based on recommendations tailored to their age group. According to WHO guidelines, vaccination schedules are as follows: a one- or two-dose schedule for girls aged 9–14 years, a one- or two-dose schedule for girls and women aged 15–20 years, or two doses with a 6-month interval for women older than 21 years [39].

HPV vaccines are typically well tolerated, with the majority of adverse effects being mild and transient. Common reactions include pain, redness, and swelling at the injection site, with pain reported in up to 80% of vaccinated individuals for both bivalent and quadrivalent vaccines. Systemic reactions, such as fever, headache, myalgia, dizziness, and nausea, are generally mild and limited in duration as well. Although post-vaccination anaphylactic reactions and syncope have been observed, they are rare and can be prevented with appropriate care [40]. HPV vaccines are contraindicated in individuals with a documented allergic reaction to the vaccines or any of their constituents, as well as in individuals with a history of immediate hypersensitivity to *Saccharomyces cerevisiae* since the protein constituents of the vaccines are produced in this yeast species [41]. Additionally, HPV vaccination is generally not recommended for pregnant women, who should defer vaccination until after pregnancy [42].

### 3.3. Varicella Zoster and Herpes Zoster

Varicella zoster virus (VZV) is the causative pathogen of varicella (chickenpox), a common and highly contagious disease characterized by an intensely pruritic vesicular rash on the trunk and face. Primary infection typically occurs in childhood and is generally benign and self-limiting, though complications such as pneumonia and encephalitis may occasionally be life-threatening [43]. Following resolution of the primary infection, VZV can establish latency in the dorsal root ganglion cells of sensory neurons or sensory ganglia of cranial nerves, with reactivation in later life causing herpes zoster (shingles) or Ramsay–Hunt syndrome [43]. Herpes zoster is characterized by a dermatomal rash with painful blistering [44]. Elderly and immunosuppressed individuals are at particular risk of VZV reactivation [45]. After resolution of the herpes zoster rash, patients may continue to suffer from pain due to postherpetic neuralgia, which has a debilitating impact on quality of life [44]. Fortunately, varicella is a vaccine-preventable disease, and since its introduction in 1995, the vaccine has been incorporated into the national immunization programs of many countries worldwide and has substantially reduced the incidence of infections [44]. Similarly, shingles can be prevented in most patients via vaccination [46].

#### 3.3.1. Vaccine Platforms

The varicella zoster vaccine contains a live, weakened form of the varicella zoster virus (VZV). Administered by subcutaneous injection, it stimulates both antibody-mediated (humoral) and cell-mediated immune responses. Specifically, the vaccine triggers the production of IgG antibodies and activates VZV-specific CD4+ T helper cells and CD8+ cytotoxic T lymphocytes [47]. The vaccine may also be administered in combination with the measles, mumps, and rubella (MMR) vaccine [48].

Two vaccines have been licensed: Shingrix, a subunit vaccine, and Zostavax, a live attenuated vaccine. Shingrix, the newer option, confers more than 90% protection, whereas Zostavax provides 50–60% protection and was discontinued in the USA in 2020 [49]. This section will focus on Shingrix. Shingrix is an inactivated vaccine that includes VZV glycoprotein E along with an adjuvant. Glycoprotein E, which is abundantly present on the surface of cells infected with zoster virus, is essential for viral replication. The adjuvant ASO1B is included to enhance the strength of the cell-mediated immune response. It is administered intramuscularly in two doses [50].

Studies have consistently shown that the varicella vaccine is effective in preventing varicella, mitigating the severity of disease, and lowering the risk of transmission [51,52,53]. A study evaluating the efficacy and safety of the varicella vaccine in Norwegian and Swedish children found that the vaccine’s efficacy against confirmed varicella was 92.1% in the two-dose group and 72.3% in the one-dose group among Norwegian children, compared to 92.6% and 58.0% in these respective groups in the Swedish cohort [51].

For Shingrix, two large-scale, multinational, randomized, placebo-controlled phase 3 trials were conducted to evaluate its efficacy and safety in adults aged 50 and older (ZOE-50) [54] and in those aged 70 and older (ZOE-70) [55]. ZOE-50 reported an incidence of herpes zoster of 0.3 per 1000 person-years in the vaccine group compared to 9.1 per 1000 person-years in the placebo group, reflecting an overall vaccine efficacy of 97.2%. Similarly, ZOE-70 reported a substantially lower incidence of herpes zoster in the vaccinated group (0.9 per 1000 person-years) compared to 9.2 per 1000 person-years in the placebo group, reflecting an overall vaccine efficacy of 89.8%. The vaccine also demonstrated 91.2% efficacy in preventing postherpetic neuralgia.

The vaccine is generally well tolerated; in the ZOE-50 and ZOE-70 trials, participants reported common local reactions such as pain (78%), redness (38.1%), and swelling (25.9%), as well as systemic reactions, including myalgia (44.7%), fatigue (44.5%), headache (37.7%), shivering (26.8%), fever (20.5%), and gastrointestinal symptoms (17.3%), within seven days of receiving both doses of the vaccine [54,55].

#### 3.3.2. Indications and Contraindications

The WHO recommends varicella vaccination on the basis that it confers public health benefits by reducing morbidity and mortality, especially among vulnerable populations such as neonates and immunocompromised individuals [56]. The vaccine is licensed for use in the protection against varicella (not herpes zoster) in individuals aged 12 months and older. In the pediatric population, the first dose is typically given between 12 and 15 months of age, and the second dose between 4 and 6 years of age [57]. It is also recommended for adolescents aged 13 and above and adults without evidence of immunity [58].

The vaccine is also approved for post-exposure prophylaxis, which substantially reduced the risk of infection (23% versus 78%) and the severity of disease in children who received the vaccine within 3 days of exposure to VZV [59]. The varicella vaccine is generally well tolerated, with local reactions such as tenderness, swelling, and redness at the injection site being the most frequently reported adverse events [51,53,60,61,62,63,64,65]. Redness after the first vaccination was reported in 20.0–31.3% of children in the Norwegian cohort and 13.8–25.6% of children in the Swedish cohort. Following the second vaccination, the most common adverse event in Norwegian children was redness at the injection site (10.0–35.5%), while pain was noted in 13.8–22.6% of Swedish children [51].

The most common systemic adverse reactions reported are fever and headache [60,61,62,63,64,65]. Though rare, serious adverse events arising from immunization, such as febrile seizures, urticaria, disseminated viral infection, and anaphylaxis, have been reported [47]. Varicella vaccine is contraindicated in pregnant women, and women are advised to use effective contraception and delay pregnancy for three months after vaccination. It is also contraindicated in individuals with a history of severe allergic reactions to a previous dose of the vaccine or a component of it, as well as those individuals who are immunosuppressed or have immunodeficiency [47].

Shingrix is recommended in older adults aged 50 and older for the prevention of herpes zoster and its complications, such as postherpetic neuralgia. It is also recommended for adults aged 18 years and above who are or will be at increased risk of herpes zoster because of immunodeficiency or immunosuppression because of disease or therapy [66]. Two doses of the vaccine are administered; the second dose is given 2–6 months after the first. In the immunosuppressed or immunodeficient, the second dose is administered 1–2 months after the first. It may be given to individuals who have had prior herpes zoster infection but should be delayed for about one year after infection. It is contraindicated in patients with a history of anaphylactic reactions to any component of the vaccine or those who have previously received a dose. Shingrix is not contraindicated in pregnant or breastfeeding women, or in severely immunocompromised patients, but is generally not recommended for use in these patients due to lack of data establishing its safety in these groups [66].

### 3.4. Melanoma

Melanoma, a form of skin cancer originating from melanocytes, is characterized by asymmetrical, irregularly shaped, brown or black lesions that change in size, shape, or color over time. If not detected early, melanoma poses considerable treatment difficulties and is associated with high mortality rates; the 5-year survival rates for patients with metastatic melanoma who underwent surgical resection were less than 20% [67]. Currently, the development of vaccines against melanoma is underway, but no vaccines have been approved yet.

#### Vaccine Platforms

Various vaccine platforms (peptides, mRNA) are currently being developed to enhance the immune response against melanoma by targeting specific tumor-associated antigens. These vaccines are intended for use as adjuvant therapy rather than prevention, aimed at patients who have had melanoma tumors surgically removed. Clinical trials have demonstrated that some of these vaccines generate robust immune responses, leading to potential improvements in treatment outcomes and a reduction in tumor recurrence.

Ott et al. (2017) demonstrated the safety and immunogenicity of a peptide-based vaccine targeting up to 20 personalized tumor neoantigens. In a phase 1 trial of six patients, four showed no recurrence 25 months post vaccination, while two with progressive disease achieved complete tumor regression following treatment with anti-PD-1 therapy. Adverse events were mild and included flu-like symptoms, injection site reactions, rash, and fatigue [68]. Similarly, a phase 1 study of mRNA-4157 (V940), an mRNA platform targeting up to 34 patient-specific tumor neoantigens, showed promising results in patients with resected cutaneous melanoma. The therapy, when used alone or in combination with an anti-PD1 antibody (pembrolizumab), induced de novo and enhanced pre-existing T-cell responses while producing no serious adverse events [69]. These findings support the development of personalized neoantigen vaccines for patients to address key challenges in cancer treatment, such as tumor heterogeneity and selective targeting of tumor while sparing healthy tissues.

An antigen-engineered dendritic cell vaccine developed by Butterfield et al. appeared to induce robust T cell responses and tumor regression or stabilization in several subjects, while resulting in little local or systemic toxicity [70]. Furthermore, a study comparing the long-term outcomes of patients with metastatic melanoma vaccinated with a multi-peptide vaccine to that of a group of unvaccinated historical controls found that median survival was significantly longer for vaccinated patients (5.4 vs. 1.3 years) [71].

More clinical evidence is required to clearly delineate the role of melanoma vaccines in conjunction with other therapeutic modalities. Current vaccine platforms for melanoma are summarized in Table 2.

### 3.5. Atopic Dermatitis

Atopic dermatitis (AD), a form of eczema, is the most common chronic inflammatory skin condition and is characterized by pruritus, dry skin, eczematous lesions, and lichenification. It is highly prevalent worldwide, affecting approximately 10–30% of children and 2–10% of adults in developed countries, with its prevalence continuing to increase. Like acne vulgaris, the exact etiology underlying AD remains unclear and is the subject of extensive research; however, it is believed that a complex interplay of genetic and environmental factors leads to disruptions in the epidermis and abnormalities of the immune system. Through the compromised epidermis, allergens and irritants can more readily penetrate the skin, leading to local and systemic inflammatory responses. This, in turn, results in redness and itching, and the development of dry, scaly patches [72]. There are currently no approved vaccines for AD.

#### Vaccine Platforms

*Staphylococcus aureus* has been identified for its potential role in the pathogenesis in AD. Early skin colonization by *S. aureus* in infants is thought to contribute to the development of clinical AD [73]. Furthermore, cutaneous *S. aureus* may exacerbate inflammation in AD lesions, leading to secondary infection and impetiginization [74]. Consequently, immunization against *S. aureus* has the potential to decrease the frequency of flare-ups and help prevent or alleviate symptoms in high-risk atopic individuals. This strategy could also reduce the reliance on broad-spectrum antibiotics for treating *S. aureus*-related AD flares, especially against the backdrop of rising antimicrobial resistance [75]. Additionally, it would avoid disrupting the microbiome, including the commensal species *S. epidermidis* and *S. hominis*, which possess anti-inflammatory and antimicrobial properties against *S. aureus* and are also being explored as possible therapeutic options for AD patients [76]. To this end, efforts are underway to develop a vaccine against *S. aureus* as a prophylactic or therapeutic approach to AD.

A study assessing the safety and immunogenicity of the *S. aureus* four-antigen (SA4Ag) vaccine concluded that it was safe and well tolerated, and quickly generated high titers of bacteria-killing antibodies in healthy adults [77]. A meta-analysis of the safety of SA4Ag and the *S. aureus* three-antigen (SA3Ag) vaccines concluded that the two vaccines have acceptable safety profiles in adults [78].

In addition, previous studies have found that the Bacillus Calmette–Guérin (BCG) vaccine may have protective effects against atopic disease [79,80,81]. Neonatal BCG vaccination might reduce the incidence of eczema by boosting T helper (Th)1 immune responses while dampening atopy-mediating Th2 responses [82,83,84]. In a 2021 study, a slight difference in 12-month eczema incidence was observed between infants who received the BCG–Denmark vaccine (32.2%) and those who did not (36.6%). This amounted to a 4.3% adjusted risk difference (aRD) and a 12% relative risk reduction (RRR). The study also found a statistically significant difference in infants born to atopic parents, with an aRD of 11.5% and RRR of 25%. Despite these findings, the authors concluded that there is insufficient evidence to recommend BCG vaccination as a preventative measure for eczema. Furthermore, the research did not find significant effects of BCG vaccination on the age of eczema onset, its severity, or the use of eczema medications [84]. In a 2018 study, clinical AD was diagnosed in 22.7% of children in the group that received the BCG vaccine and 25.4% of children in the control group. Among children with atopic predisposition, neonatal BCG reduced the cumulative incidence of AD by 16% [85].

In summary, while there are currently no approved vaccines for AD, targeting *S. aureus* and the BCG vaccine are potential strategies for the management of AD. Vaccine platforms for atopic dermatitis are summarized in Table 3.

### 3.6. Warts

Cutaneous warts are a common group of hyperkeratotic conditions caused by HPV, which include common warts (*verruca vulgaris*), flat warts (*verruca plana*), and plantar warts (*verruca plantaris*). Their prevalence is estimated to be 7–12% [86]. These lesions are generally benign, though there is a potential for malignant transformation in rare cases [87].

At present, several treatment options are available, including topical keratolytics, cryotherapy, electrocautery, chemical cautery, and laser ablation. However, these modalities can be painful and time-intensive, and none is regarded as the definitive standard of care due to the risk of scarring, disfigurement, and recurrence [88]. Immunotherapy is emerging as a potentially effective and well-tolerated treatment modality.

#### Vaccine Platforms

To date, several studies have explored the safety and efficacy of various intralesional therapies, including injection of *Candida* antigen (an extract of *C. albicans* antigens, with immunomodulatory activity), MMR vaccine, and BCG vaccine.

A comparative study of the efficacy and safety of an intralesional *Candida* antigen injection versus topical diphencyprone (DPCP, a contact allergen used as topical immunotherapy) in the treatment of warts found that both treatment modalities achieved wart clearance or improvement without a statistically significant difference; however, the *Candida* antigen was more effective in clearing adjacent untreated warts and had fewer adverse side effects. Among the patients (*n* = 40), 17 (42.5%) achieved complete clearance of the central treated wart; this was higher in the *Candida* group (12/20; 60%) than in the DPCP group (5/20; 25%), though this difference was not statistically significant. Warts adjacent to the treated mother wart were seen in 26 participants. Among them, 30% (3/10) in the *Candida*-treated group achieved clearance of the adjacent warts, while none (0/16) in the DPCP group did; this difference was statistically significant. Furthermore, 8 patients in the *Candida* antigen group developed side effects, compared to 20 patients (100%) in the DPCP group; this difference was also statistically significant. Nonetheless, adverse effects in both groups were transient, mainly consisting of mild edema and erythema [89].

A 2021 study compared the effectiveness and safety of intralesional MMR vaccine administration versus intralesional *Candida* antigen in the treatment of common and plantar warts. The results indicated that complete responses were achieved in 23 patients (67.7%) in the MMR group and 25 patients (73.5%) in the *Candida* antigen group, with no statistically significant difference in overall therapeutic outcomes. Both treatments were associated with mild adverse effects such as pain during injection, localized erythema and edema at the site of injection, and flu-like symptoms. Notably, none of the patients developed an infection during the 6-month follow-up period [90].

In contrast, another 2021 study specifically examined the efficacy of intralesional MMR vaccine versus *C. albicans*-specific antigen for plantar warts, finding a significant difference in cure rates. The *C. albicans* antigen demonstrated an impressive 80.0% cure rate, compared to just 26.7% for the MMR vaccine. Moreover, patients treated with the *C. albicans* antigen required fewer sessions on average (3.98 sessions) than those receiving the MMR vaccine (4.24 sessions), indicating variability in the MMR vaccine’s efficacy. Both treatments were generally well tolerated; side effects included mild and transient pain, redness, and swelling. No recurrences were observed in cured patients during the 2-month follow-up period [91].

In addition, a separate 2021 study evaluated the efficacy and safety of three antigens—MMR vaccine, *Candida* antigen, and purified protein derivative (PPD)—in the treatment of periungual warts. This study included 150 patients who were randomly assigned to three groups, with 50 patients in each. Each agent was injected intra-lesionally at a dose of 0.1 mL into the largest wart at 2-week intervals until complete clearance or for a maximum of five sessions. Complete clearance of warts was observed in 70% in the PPD group, 80% in the *Candida* antigen group, and 74% in the MMR vaccine group. There was no statistically significant difference in therapeutic response among the three groups. Adverse effects were transient and insignificant, consisting mainly of tolerable pain at the injection site, erythema, edema, and flu-like symptoms. No recurrences were reported in any of the groups [92].

Building on these findings, a 2022 study assessed the safety and effectiveness of intralesional injections of MMR antigen, BCG vaccine, and *Candida* antigen for treating multiple warts. This study reported complete wart clearance rates of 73.3% for the MMR group, 70.0% for the BCG group, and 43.3% for the *Candida* antigen group. While infrequent adverse effects such as pain, erythema, and slight induration were noted across all groups, they did not warrant cessation of treatment [93].

Ostensibly, there is variability in responses to immunotherapy and the potential for different intralesional therapies in managing wart conditions.

Vaccine platforms for warts are summarized in Table 4.

### 3.7. Mucocutaneous candidiasis

Mucocutaneous candidiasis is a fungal infection of the skin and mucous membranes caused by a species of *Candida*, most commonly *Candida albicans* (*C. albicans*). It commonly manifests as a well-demarcated erythematous patch with confluent papules or on the mucosa as white plaques [94]. Epidemiologically, mucocutaneous candidiasis is widespread; vulvovaginal candidiasis has an estimated global annual prevalence of approximately 3.8%, affecting more than 370 million women who experience at least one episode during their lifetimes [95]. In particular, recurrent vulvovaginal candidiasis (RVVC) considerably impairs quality of life due to its symptomatology, frequency, and unpredictability [96].

To date, no licensed vaccines or immunotherapeutic strategies are available for the treatment of RVVC; current management is based on repeated or extended courses of antifungal therapy, which may vary in its effectiveness, pose safety concerns [97], and contribute to antifungal resistance [98]. Thus, a safe and effective vaccine would be a valuable addition to the strategies for combating RVVC.

#### Vaccine Platforms

A randomized, double-blind, placebo-controlled trial of an active *Candida* vaccine (NDV-3A) containing a recombinant *C. albicans* adhesin/invasin protein for the prevention of RVVC was conducted in 188 women. The results indicated that a single intramuscular dose of NDV-3A was safe and elicited strong B- and T-cell immune responses. Notably, at 12 months post vaccination, 42% of vaccinated patients were symptom-free versus 22% in the placebo group, a difference that was statistically significant. Moreover, in the vaccinated group, the median time to the first symptomatic episode was substantially longer at 210 days compared to 105 days in placebo in a subset of patients under 40 (*n* = 137). The most common adverse events were injection site reactions like pain, tenderness, and redness. While some treatment-emergent systemic adverse events such as urinary tract infection were reported in both the placebo and the vaccinated group, almost all were mild or moderate and not related to study treatment, and there was no significant difference in the rates of adverse events among the groups. Overall, NDV-3A demonstrated significant potential in reducing symptomatic episodes in younger women [99].

### 3.8. Buruli ulcer

Buruli ulcer (BU) is a skin-related neglected tropical disease (NTD) caused by *Mycobacterium ulcerans* (*M. ulcerans*), characterized by chronic, non-healing skin ulcers and necrosis. This infection has the potential to cause permanent disfigurement and has been increasingly reported in regions such as Africa, South America, and the Western Pacific [100,101]. At present, there is no clinically approved vaccine specifically for *M. ulcerans*; the live attenuated BCG vaccine is the only available option that possibly confers protection.

#### Vaccine Platforms

While the BCG vaccine is primarily used for protection against *Mycobacterium tuberculosis* (*M. tuberculosis*), it is also potentially protective against *M. ulcerans* and *Mycobacterium leprae* (*M. leprae*) due to the cross-reactivity of B and T cells to antigens common among different mycobacterial species [102].

Nonetheless, evidence of BCG immunization against *M. ulcerans* infection remains inconclusive; some studies suggest it confers partial, transient protection or limits disease progression, particularly reducing severe outcomes like osteomyelitis, while others find no correlation between immunization and disease status.

In a trial conducted in Uganda, BCG vaccination provided 47% protection against BU, mainly limited to the first year among those with tuberculin reactions of less than 4 mm before vaccination. While BCG conferred no additional benefit for those with prior disease or an existing BCG scar, these groups showed high protection (88% and 82%, respectively). Furthermore, lesions that developed were smaller in vaccinated individuals than in the unvaccinated. While these findings suggest that BCG vaccination confers transient protection against *M. ulcerans*, its long-term effects remain uncertain [103].

BCG vaccination at birth has also been found to protect BU patients against severe disseminated disease, such as osteomyelitis. In a study evaluating the prophylactic effect of BCG vaccination, only 12.2% of osteomyelitis patients had a BCG scar, compared to 35.7% without [104]. However, a case-control study in the Democratic Republic of Congo, Ghana, and Togo found no significant link between a BCG scar and Buruli ulcer status, suggesting insufficient evidence that routine BCG vaccination protects against BU [105].

Vaccine platforms for Buruli ulcer are summarized in Table 5.

### 3.9. Leprosy

Leprosy, or Hansen disease, is a skin-related NTD caused by chronic infection by *Mycobacterium leprae* (*M. leprae*). It is characterized by hypopigmented skin lesions and peripheral nerve damage [106]. At present, there is no clinically approved vaccine specifically for *M. leprae*; the live attenuated BCG vaccine is the only available option that possibly confers protection.

#### Vaccine Platforms

In a 2019 study, it was reported that individuals with two BCG scars had a significantly lower risk of developing leprosy compared to those without prior vaccination, and those with two scars took more time on average to develop the disease [107]. In a 2020 study, pediatric patients who received the BCG vaccination had a higher ratio of paucibacillary (PB) to multibacillary (MB) leprosy (5.3:1) compared to the unvaccinated group (1.2:1), with a statistically significantly higher proportion of MB leprosy in the unvaccinated group, suggesting that BCG vaccination may potentially reduce the severity of leprosy [108]. Overall, the effectiveness of BCG vaccination in protecting against leprosy was reported to vary between 20% and 90% [109].

Fortunately, at least two new vaccines are currently in development: the *Mycobacterium indicus pranii* (MIP) vaccine and the LepVax subunit vaccine. The former is an inactivated, non-tuberculous mycobacterial vaccine used as an adjunct immunotherapy for MB leprosy patients [110] and has been shown to reduce bacillary load and enhance treatment outcomes in leprosy patients without any adverse effects [111]. The latter was developed based on an *M. leprae* recombinant polyprotein to provide both pre-exposure and post-exposure protection against *M. leprae* infection. It has been shown to be safe and mitigate nerve damage caused by *M. leprae* infection, though testing of the vaccine is still ongoing [112].

Vaccine platforms for leprosy are summarized in Table 6.

### 3.10. Leishmaniasis

Leishmaniasis is a parasitic NTD caused by protozoans of the *Leishmania* genus, which are transmitted by female phlebotomine sandflies [113]. It can manifest in cutaneous, mucosal, or visceral forms. Cutaneous leishmaniasis (CL) is characterized by erythematous macules or papules that develop central ulceration and atrophic scars. It has an annual incidence of over 1.5 million cases in the tropical and subtropical regions [114]. Visceral leishmaniasis (VL), also known as kala-azar, is the most severe form of leishmaniasis, characterized by fever, hepatosplenomegaly, and pancytopenia, and is responsible for over 20,000 deaths annually [114]. Post-kala-azar dermal leishmaniasis (PKDL) is a chronic skin condition characterized by hypopigmented lesions that can occur following clinical cure from VL [115].

Current systemic treatment options include amphotericin B, pentavalent antimonial compounds, pentamidine, miltefosine, and azole antifungals; topical options include paromomycin [116]. However, these therapies come with significant limitations, such as toxicity, numerous adverse effects, and challenges with patient adherence. Vaccination is viewed as the most effective strategy for controlling the disease [117]. Furthermore, the progression of VL and PKDL is linked to inadequate cell-mediated immunity, including CD8+ T cell responses, which limits the efficacy of anti-leishmanial drugs [118]. To address this, the development of both prophylactic and therapeutic vaccines has been identified as a critical need. Although there are currently no licensed vaccines for CL, several candidates are undergoing clinical trials.

#### Vaccine Platforms

For decades, leishmanization, which involved the intradermal inoculation of live *Leishmania major* (*L. major*) parasites, was practiced in the Middle East. Although it provided up to 90% protection against re-infection, it is no longer practiced due to safety and ethical concerns, as the lesions at the site of inoculation could last for months in some individuals [119]. Nonetheless, empirical evidence continues to support the potential of a safer leishmanization approach for effective protection against CL [120].

Since then, leishmaniasis vaccine development has progressed through three generations. First-generation vaccines include whole-killed or live attenuated *Leishmania* parasites. While these vaccines have demonstrated some efficacy, they are often limited by safety concerns. Second-generation vaccines have shifted towards recombinant *Leishmania* antigens produced using genetically engineered viruses or bacteria. Third-generation vaccines aim to improve the precision of vaccination through nucleic acid-based formulations such as DNA or RNA [121].

Leishvaccine, a first-generation vaccine containing whole-killed *L. amazonensis*, was evaluated through a cluster-randomized trial conducted from 2002 to 2011 across 108 localities in a CL-endemic region of southeast Brazil. Analysis showed a significant reduction in CL incidence in the vaccinated area compared to the placebo area [122]. Today, advancements in CRISPR-based genome editing have allowed for the creation of safer attenuated *Leishmania* strains that induce protective immunity. Notably, a recent study showed that mice immunized with centrin gene-deleted *Leishmania* parasites mounted strong immune responses and did not develop visible lesions, comparable to traditional leishmanization [119]. In addition, US Food and Drug Administration (FDA)-approved treatments, such as radiofrequency-induced heat therapy, can help to mitigate risks of adverse reactions [123].

Second-generation vaccines can attain high levels of purity, which facilitates standardized, large-scale, and cost-effective production. Many second-generation vaccines are designed primarily for canine populations, as dogs are key reservoir hosts for *Leishmania* species. Several second-generation vaccines for human use have also been studied. One such example is LEISH-F3, a vaccine that incorporates a recombinant fusion protein delivered with strong Th1-inducing adjuvants, which effectively induced strong immune responses against VL in healthy adults. The vaccine was well tolerated; the most common adverse effects reported were local tenderness (95.8%) and fatigue (50.0%); no serious adverse events were reported [124]. Other examples include LEISH-F1, LEISH-F2, and Poly-T Leish [121]. However, despite extensive animal studies, few candidates have progressed to advanced human or non-human primate trials.

The first DNA antileishmanial vaccine contained the gene encoding gp63; this vaccine conferred protection against *L. major* [125] and *L. mexicana* in murine models [126].

Another promising third-generation vaccine is ChAd63-KH, an adenovirus-based vaccine that contains a gene encoding two *L. donovani* proteins: kinetoplastid membrane protein 11 (KMP-11) and hydrophilic acylated surface protein B (HASB). A phase 1 clinical trial of ChAd63-KH conducted in 20 healthy volunteers from the UK found that it was safe and induced robust CD8+ T cell responses in all vaccinated subjects; adverse events were largely mild injection site reactions [118]. A phase 2a safety and immunogenicity trial of ChAd63-KH conducted in Sudanese PKDL patients found that it elicited strong immune responses and minimal adverse reactions, including local itch, swelling, and pain. Clinically, 30.4% of patients experienced more than 90% improvement, while 21.7% experienced partial improvement. A randomized controlled trial is ongoing to confirm its clinical efficacy [127]. Other notable examples that have been found to elicit strong Th1 immune responses include the HisAK70 vaccine, which encodes seven *Leishmania* antigens [128], and LeishDNAvax, a naked multi-epitope DNA vaccine [129]. Though DNA vaccines are advantageous in terms of stability, cost-effectiveness, and long-lasting antigen production [130], there are concerns about safety, including regarding genomic integration [131]. RNA vaccines are still in the early stages, though a recent study demonstrated promising results in providing protection against *L. donovani* [132].

To summarize the current state of *Leishmania* vaccine development, creating an effective vaccine is contingent upon a comprehensive understanding of host–pathogen interactions, optimal vaccine candidates, and suitable adjuvants or delivery methods. Success hinges on inducing strong, long-lasting Th1 memory responses and IFN-γ-producing T cells. However, human trials have struggled with sustained immunity due to the parasites’ antigen diversity, varied immune responses, and the potential effects of sandfly saliva. Each type of vaccine faces distinct hurdles: killed vaccines provide limited protection, live attenuated vaccines pose safety risks, and nucleic acid-based vaccines are still in development, with lingering safety concerns. Financial disincentives further impede progress, as leishmaniasis predominantly affects developing regions, complicating cost recovery for research and trials. Despite these challenges, global efforts are ongoing, with several vaccine candidates undergoing clinical trials. While thus far no human vaccine has been approved, animal vaccination also plays a crucial role in controlling transmission, and with increased resources, the goal of an effective vaccine remains in sight.

Beyond vaccines, research has also been conducted on a range of immunotherapies, including recombinant cytokines, cytokine antagonists, immune checkpoint inhibitors, TLR agonists, cellular receptor and signaling modulators, enzyme inhibitors, anti-inflammatory and antioxidant agents, and cellular therapy [133].

Vaccine platforms for leishmaniasis are summarized in Table 7.

### 3.11. Smallpox

Smallpox is a highly infectious disease transmitted via the respiratory route, and is characterized by fever, rash, and headache [134]. Prior to the development of the smallpox vaccine, it ravaged communities worldwide, causing high mortality rates and significant scarring in survivors. The global smallpox eradication campaign, declared successful in 1980 by the WHO, remains one of the greatest achievements in public health. The smallpox vaccine, first developed by Edward Jenner in the 18th century, was the first successful vaccine and laid the foundation for the development of all subsequent vaccines. However, in recent years, concerns about its potential re-emergence—particularly through bioterrorism—have led to ongoing vigilance and preparedness efforts, despite the virus having been eradicated from natural populations and routine vaccinations having ceased [135].

#### 3.11.1. Vaccine Platforms

Generally, the first-generation vaccines, introduced in the 1950s, were less safe and reliable, as they contained live, non-attenuated vaccinia viruses derived from live animals. Therefore, their use posed the risk of adverse side effects, such as acute vaccinia syndrome and myopericarditis, as well as unintended viral transmission. Second-generation live virus vaccines produced using tissue or cell line cultures and introduced in the 2000s had fewer severe side effects compared to the first generation but still posed a risk for serious adverse effects, such as encephalitis, especially in vulnerable populations. In contrast, the third-generation attenuated vaccines had lower virulence and replication abilities and thus better safety profiles. Recipients of the third-generation vaccines generally experienced minor side effects like fatigue, fever, and lymphadenopathy, rendering them suitable for a wider range of individuals, including the immunocompromised [136].

Currently, two smallpox vaccines are licensed for use in the USA by the FDA: ACAM2000, a second-generation vaccine approved in 2007, and JYNNEOS (also known as Modified Vaccinia Ankara-Bavarian Nordic [MVA-BN], Imvamune, or Imvanex), a third-generation non-replicating attenuated vaccine approved more recently in 2019. A third vaccine, the Aventis Pasteur Smallpox Vaccine (APSV), is an investigational vaccine that may be used in a smallpox emergency [137].

A phase 2 trial evaluated the safety and immunogenicity of one high dose (HD) versus two standard doses (SD) of the MVA vaccine. Both regimens were well tolerated, and seroconversion occurred earlier in the HD group; however, the median time to seroconversion was similar (14 days). Although the HD peak titer was superior to one SD dose, it was inferior to the two-dose SD regimen, suggesting the standard two-dose approach has greater immunogenicity [138].

A phase 1 clinical trial investigated the safety and immunogenicity of the MVA vaccine in 15 healthy individuals and 45 subjects with allergic rhinitis, a history of atopic dermatitis (AD), or mild active AD. MVA was well tolerated across all groups, with no serious adverse events, clinically relevant skin reactions, or notable differences in adverse reactions between healthy and AD subjects. By the second dose, 100% of vaccinees had seroconverted, with robust antibody responses, suggesting MVA has similar safety and immunogenicity profiles in both healthy and AD populations [139].

Another study evaluated a shortened dosing schedule (days 0 and 7) of IMVAMUNE compared to the standard schedule (days 0 and 28). Overall, IMVAMUNE was safe and well tolerated, and elicited strong immune responses, though the standard schedule elicited significantly greater antibody titers and a higher proportion of responders [140].

A phase 3 trial involving 440 participants compared the two licensed vaccines and found that MVA induced significantly higher neutralizing antibody titers than ACAM2000 at peak visits (geometric mean titers of 153.5 vs. 79.3, respectively) and demonstrated similar seroconversion rates after a single dose (90.8% vs. 91.8%, respectively). In addition, the MVA vaccine was associated with fever adverse events of grade 3 or higher, and no safety concerns were identified [141].

CJ-50300, a second-generation cell culture-derived smallpox vaccine developed in South Korea, was evaluated in a phase 3 clinical trial involving 145 healthy adults previously vaccinated against smallpox. Among the 139 participants who completed the study, 95.0% exhibited cutaneous “take” reactions and 88.5% demonstrated humoral immunogenicity. Although 95.9% reported vaccine-related adverse events, no serious reactions were observed [142].

#### 3.11.2. Indications and Contraindications

In immunocompromised individuals, such as those with cancer, organ transplant recipients, HIV-infected individuals, and those on immunosuppressive therapy, not all smallpox vaccines are safe due to the potential for severe side effects. For instance, the ACAM2000 vaccine is contraindicated for individuals with severe immunodeficiency due to the risk of severe vaccinia skin infections. Similarly, the LC16 vaccine is not recommended for those with severe immunodeficiency or those receiving immunosuppressive treatment [143]. In contrast, the MVA-BN vaccine has a favorable safety profile and immunogenicity in studies involving hematopoietic stem cell transplant recipients and individuals with HIV, making it a safer option for these groups [144,145].

The use of smallpox vaccines in pregnant women and newborns remains a subject of debate. The ACAM2000 vaccine has been linked to a case of mother-to-child transmission of the vaccinia virus through breastfeeding [146]; consequently, the CDC prohibits its use in pregnant and breastfeeding women, as well as in infants under one year old [147]. In contrast, the JYNNEOS vaccine has shown no evidence of fetal malformations or developmental delays in animal studies [148], though its safety in pregnant women remains under investigation. Global consensus on the appropriate timing for vaccinating newborns against smallpox is still lacking, as the risk of adverse reactions in infants under one year old is high.

Vaccine platforms for smallpox are summarized in Table 8.

### 3.12. Mpox

Mpox is a zoonotic infectious disease caused by the monkeypox virus, endemic in certain regions of Africa. Recent outbreaks have sparked global concern; in August 2024, the WHO declared the outbreak in the Democratic Republic of Congo and the growing number of affected countries a Public Health Emergency of International Concern. By the end of May 2023, more than 87,545 cases and 141 deaths had been reported, the majority of which occurred in non-endemic countries, largely as a result of human-to-human transmission [149]. In most patients, mpox typically manifests with skin lesions that progress from erythematous macules and papules to vesicles and deep-seated, dome-shaped pustules, which may become umbilicated [150].

#### Vaccine Platforms

Several vaccines are available for preventing mpox, including traditional smallpox vaccines, which have demonstrated cross-protective immunity due to shared immunological markers among orthopoxviruses. Historical data from animal studies in the 1960s revealed that antibodies generated by smallpox vaccines could recognize and neutralize numerous orthopoxvirus proteins, conferring cross-immunity against mpox [151].

In China, individuals who received the first-generation live smallpox vaccine Tian Tan before 1981 were found to have maintained virus-specific antibodies that provided partial protection against mpox for over 42 years [152]. This long-lasting immunity is corroborated by evidence showing that smallpox vaccines, including modern versions, are effective for both pre- and post-exposure prophylaxis [153]. In England, MVA-BN demonstrated 78% effectiveness in preventing symptomatic mpox 14 days after pre-exposure immunization [154]. In addition, during the 2022 outbreak, post-exposure use of one dose of MVA-BN demonstrated an adjusted effectiveness of 88.8% [155].

Although smallpox vaccines play a pivotal role in combatting mpox, there is a growing need for safer and more efficacious vaccines specifically targeting MPXV, as smallpox vaccines are not entirely satisfactory in terms of their effectiveness and safety profiles [136]. The development of these vaccines has been met with challenges, including slow progress in animal studies, a lack of clinical trial data, and the evolving nature of MPXV strains. Recent research has identified promising vaccine targets, including immunogens such as L1R, B5R, A27L, and A33R, which can enhance vaccine efficacy. Furthermore, intracellular mature virions (IMVs) and extracellular enveloped virions (EEVs) are also being explored as targets for new vaccine development [156].

The 4pox DNA vaccine targeting immunogenic sites L1, A27, B5, and A33 confers comparable protection to traditional smallpox vaccines like ACAM2000 and MVA-BN. It does not require drug delivery systems like mRNA vaccines and has demonstrated protective immunity in animal models against orthopoxviruses [157,158]. However, it has yet to undergo clinical trials in humans. Multivalent mRNA vaccines designed to target multiple MPXV antigens have also shown promise in generating T-cell responses and protecting against vaccinia virus in animal models [159]. This indicates that combining antigens from both mature and enveloped virions enhances protection [160].

Additionally, protein-based subunit vaccines containing antigenic components such as envelope proteins from mature and enveloped virions have been developed. Although they have shown efficacy in animal studies with adjuvants [161], they are still in the pre-clinical stage and have not yet undergone clinical trials in humans.

Vaccine platforms for mpox are summarized in Table 9.

### 3.13. Hand, Foot, and Mouth Disease (HFMD)

HFMD is a highly contagious viral infection most frequently caused by coxsackie A virus (CVA) and enterovirus A71 (EV-A71). It classically presents with oral ulcers and maculopapular or vesicular rash on the hands and feet, accompanied by non-specific symptoms such as fever and malaise. The disease most commonly affects children under the age of 10. Symptoms usually resolve within several days without lasting complications. Currently, there are no specific pharmacological treatments for HFMD; management is primarily symptomatic. However, vaccination remains the most effective strategy for controlling the disease [162].

#### Vaccine Platforms

Currently, both monovalent and polyvalent vaccines against the HFMD pathogen are available. The inactivated EV-A71 vaccine, a monovalent vaccine approved by the China Food & Drug Administration, demonstrated high efficacy rates of between 94.7% and 97.4% in several phase 3 trials [163,164,165]. In one trial, the vaccine was 100% effective in preventing EV71-associated hospitalization and neurological complications [164]. A phase 4 study involving 155,995 children (of whom 40,724 received 2 doses of the vaccine) demonstrated an overall effectiveness of 89.7% against EV71 infection and a 4.58% rate of reported adverse events over a 14-month follow-up period [166]. A comprehensive safety assessment indicated a favorable safety profile, with fever and pain at the injection site being the most common systemic and local reactions, respectively [167]. However, monovalent vaccines are limited in scope, as they only protect against a single genotype. To widen the net of protection, polyvalent vaccines combining different serotypes or using innovative chimeric constructs have been developed. These vaccines, including bivalent (CVA6 and CVA10), trivalent (EV71, CVA6, and CVA16; CVA6, CVA10, and CVA16), and tetravalent (EV71, CVA6, CVA10, and CVA16) formulations, have shown promising neutralizing antibody responses and cell-mediated immune responses in animal studies [168,169,170,171]. In addition, virus-like particles, chimeric vaccines, and recombinant vector vaccines have demonstrated broad protective effects and systemic immune responses [172]. Further development of polyvalent vaccines may include advancements in peptide-based and DNA/RNA vaccine technologies [173,174].

Vaccine platforms for HFMD are summarized in Table 10.

### 3.14. Group B Streptococcus (Streptococcus agalactiae)

Group B *Streptococcus* (GBS) is a pathogen responsible for invasive infections with a wide spectrum of manifestations, particularly in vulnerable populations such as newborns, pregnant individuals, and the elderly. Skin and soft tissue infections are among the most common manifestations of GBS infection; these include cellulitis, abscesses, foot infections, pressure ulcers, and, more rarely, necrotizing fasciitis [175]. Rarely, GBS infection may progress to streptococcal toxic shock syndrome (STSS), a life-threatening illness characterized by acute sepsis, shock, multiorgan failure, and a high fatality rate [175]. In non-pregnant adults, the incidence of STSS from invasive GBS has increased, with penicillin- and clindamycin-resistant strains detected [176]. Despite advances in preventive measures like intrapartum antibiotic prophylaxis, GBS continues to pose a global health challenge and remains a significant contributor to neonatal morbidity worldwide. While antibiotic therapy is the mainstay of treatment, rising antimicrobial resistance has driven vaccination development strategies to combat this threat [177]. Indeed, the WHO has identified GBS as one of 17 high-priority global endemic pathogens for vaccine research and development [178]. It is estimated that a GBS vaccine with 80% efficacy and 90% maternal coverage could avert 107,000 infant deaths and stillbirths [179].

#### Vaccine Platforms

Current vaccine development focuses on vaccinating mothers in the third trimester to prevent neonatal and maternal GBS disease. A phase 1b/2 trial of an investigational trivalent GBS vaccine targeting capsular polysaccharides (CPSs) Ia, Ib, and III assessed its safety and immunogenicity in non-pregnant and pregnant women, along with antibody transfer to infants. The vaccine induced significantly higher serotype-specific antibody concentrations in both groups compared to the placebo. Reactogenicity was mild: the most reported local adverse reactions included pain, headache, fatigue, myalgia, and warmth; no severe adverse events or deaths were linked or attributed to vaccination. Maternal vaccination resulted in higher antibody concentrations in infants [180]. Several other trials have also found that the trivalent vaccine had a favorable safety profile and robust immunogenicity [181,182,183,184]. Furthermore, a trial of a second dose found that administering the second dose 4–6 years after the first was immunogenic, particularly benefitting women with undetectable baseline anti-GBS concentrations, and safe, with mild adverse events such as pain, headache, and fatigue [185]. However, the vaccine may be less efficacious in HIV-infected pregnant women, resulting in lower antibody responses and reduced maternal antibody transfer to infants, potentially compromising neonatal protection against GBS [186].

A phase 1/2 trial evaluating the hexavalent CPS-cross-reactive material 197 glycoconjugate vaccine (GBS6) targeting CPS serotypes Ia, Ib, and II through V found that it induced strong immune responses and was well tolerated in healthy adults. Pain at the injection site, fatigue, and headache were among the most frequently reported solicited reactions but were mild and transient. [187]. Another phase 2 trial evaluating the same vaccine found it to be immunogenic and safe: maternal vaccination produced serotype-specific anti-CPS antibodies, which were transferred to infants at maternal-to-infant ratios of 0.4 to 1.3. A percentage of 57–97% of infants developed antibodies that were associated with a reduced risk of disease. No significant safety concerns related to the trial vaccine were identified; adverse event rates were comparable to those in the placebo group, consisting primarily of pain at the injection site, headache, and vomiting [188].

In a study on STSS cases in non-pregnant individuals in Japan, the hexavalent conjugate vaccine demonstrated 91.4% efficacy in preventing STSS, significantly outperforming the trivalent conjugate vaccine, which showed 61.2% efficacy. The hexavalent vaccine’s superior coverage can be attributed to the inclusion of CPS serotype V, the second most frequently isolated serotype in STSS cases. These findings suggest that vaccinating non-pregnant adults with the hexavalent conjugate vaccine could reduce GBS-related STSS cases by over 90% and, consequently, improve public health outcomes [176].

Other vaccine candidates include GBS-NN, consisting of fused N-terminal domains of the AlphaC and Rib surface proteins. GBS-NN was found in a phase 1 trial to be well tolerated, with mostly mild injection site reactions. The use of an adjuvant significantly boosted antibody responses and achieved high IgG concentrations; these antibody levels remained elevated for up to one year [189]. The DNA vaccine SL/pVAX1-sip containing a DNA fragment encoding a portion of the surface immunogenic protein (Sip) of *S. agalactiae* was found to be safe and effective in tilapias, inducing protective immune responses without integrating into the fish’s chromosomes [190]. A GBS type III CPS–tetanus toxoid (GBS III-TT) vaccine was tested for its safety and efficacy in preventing GBS colonization in non-pregnant women. Vaccine efficacy was 36% for delaying vaginal GBS III acquisition and 43% for rectal acquisition. Vaccine recipients showed a four-fold increase in serum serotype-specific IgG. The vaccine was generally well tolerated, with only mild adverse events reported, such as tenderness or pain at the injection site, as well as systemic symptoms such as headache, malaise, and myalgia. There was no significant difference in rates of unsolicited adverse events between the vaccine groups [191].

Vaccine platforms for GBS are summarized in Table 11.

## 4. Conclusions

Notable progress has been made in elucidating the immunopathogenic processes underlying dermatological diseases, paving the way for the development of targeted vaccines for preventative and therapeutic purposes. In line with the goals of its Immunization Agenda 2030 (IA2030), the WHO has identified a new list of 17 high-priority global endemic pathogens for vaccine research and development, of which skin pathogens account for at least eight [178], reinforcing the need for continued focus in this area. While recent advances have led to the development of promising vaccine candidates, there is still plenty of room for improvement. Challenges remain in standardizing outcome measures for clinical trials, ensuring vaccine safety across diverse populations, and translating results from animal models into human subjects. Continued research is required to refine these vaccines, address existing gaps, and translate these innovations into effective clinical treatments.

## Figures and Tables

**Table 1 vaccines-13-00125-t001:** Vaccine platforms for acne vulgaris.

Therapy	Study Type	Outcomes	Reference
Inactivated *C. acnes*	Animal	Induced ear inflammation	[11]
Sialidase	Animal	Increased IL-8 production, reduced ear swelling, decreased MIP-2 release	[13]
CAMP factor	Animal	Decreased *C. acnes*-induced inflammation	[15]
	Animal	Decreased MIP-2 release	[17]
	Ex vivo	Decreased IL-8 and IL-1b production	
HylA	Animal	Decreased HylA-induced inflammation	[18]
Unspecified *C. acnes* recombinant proteins	Human	Pending	[6]
Unspecified mRNA	Human	Pending	[7]

**Table 2 vaccines-13-00125-t002:** Vaccine platforms for melanoma.

Therapy	Study Type	Outcomes	Reference
20 personalized tumor neoantigens	Human (phase 1)	Protected against recurrence in 4 out of 6 patients at 25 months post vaccination	[68]
mRNA-4157 (V940)	Human	Induced de novo and strengthened pre-existing T-cell responses to targeted neoantigens	[69]
Antigen-engineered dendritic cell vaccine	Human	Induced antigen-specific CD8+ and CD4+ T-cell responses	[70]
Multipeptide vaccine	Human	Increased median survival	[71]

**Table 3 vaccines-13-00125-t003:** Vaccine platforms for atopic dermatitis.

Therapy	Study Type	Outcomes	Reference
SA4Ag (CP5, CP8, rmClfA, rMntC)	Human	Safe; generated high titers of *S. aureus*-specific antibodies	[77]
SA4Ag and SA3Ag	Human	Acceptable safety profiles	[78]
BCG-Denmark	Human	Decreased incidence of AD	[84]
BCG	Human	Decreased incidence of AD	[85]

**Table 4 vaccines-13-00125-t004:** Vaccine platforms for warts.

Therapy	Study Type	Outcomes	Reference
Intralesional *Candida* antigen versus topical diphencyprone	Human	Intralesional *Candida* antigen more efficacious; both safe	[89]
Intralesional MMR vaccine versus *C. albicans*-specific antigen	Human	Similar rates of clearance; both safe	[90]
	Human	MMR vaccine more efficacious; both safe	[91]
Intralesional MMR vaccine versus *Candida* antigen and PPD	Human	Similar rates of clearance; all safe	[92]
Intralesional MMR vaccine versus BCG vaccine and *Candida* antigen	Human	Higher rates of clearance using MMR and BCG than *Candida*; all safe	[93]

**Table 5 vaccines-13-00125-t005:** Vaccine platforms for Buruli ulcer.

Therapy	Study Type	Outcomes	Reference
BCG vaccine	Human	47% protection; decrease in lesion size	[103]
	Human	Decreased risk of progression to osteomyelitis	[104]
	Human	No significant link	[105]

**Table 6 vaccines-13-00125-t006:** Vaccine platforms for leprosy.

Therapy	Study Type	Outcomes	Reference
BCG vaccine	Human	Decreased risk of disease; increased disease-free period	[107]
	Human	Decrease in severity	[108]
MIP vaccine	Human	Reduced bacillary load; enhanced treatment outcomes	[111]
LepVax subunit vaccine	Human	Safe; reduced risk of neuropathy	[112]

**Table 7 vaccines-13-00125-t007:** Vaccine platforms for leishmaniasis.

Therapy	Study Type	Outcomes	Reference
Live *L. major* parasites	Human	Up to 90% protection; unsafe	[119]
Leishvaccine (whole-killed *L. amazonensis*)	Human	Reduced CL incidence	[122]
Centrin-deleted *Leishmania*	Animal	Immunized mice had no visible lesions following challenge with *L. major*-infected sandflies.	[119]
LEISH-F3	Human (phase 1)	Induced strong immune responses against VL	[124]
gp63	Animal	Protected against *L. major*	[125]
	Animal	Protected against *L. mexicana*	[126]
ChAd63-KH (KMP-11 and HASB)	Human (phase 1)	Safe; induced robust CD8+ T cell responses in 100% of subjects	[118]
	Human (phase 2a)	Induced robust immune responses; minimal adverse reactions; 30.4% experienced >90% improvement; 21.7% experienced partial improvement	[127]
HisAK70 (H2A, H2B, H3, H4, A2, KMP11, and HSP70)	Animal	Induced robust immune responses, including higher iNOS/arginase activity, IFN-γ/IL-10, IFN-γ/IL-4, and GM-CSF/IL-10 ratios	[128]
LeishDNAVax (naked multi-epitope DNA vaccine)	Animal	Induced T-cell-based immunity and conferred protective effects against infection	[129]
RNA vaccine	Animal	Protected against challenge with *L. donovani*	[132]

**Table 8 vaccines-13-00125-t008:** Vaccine platforms for smallpox.

Therapy	Study Type	Outcomes	Reference
MVA	Human (phase 2)	One high dose is superior to one standard dose but inferior to two standard doses.	[138]
MVA	Human (phase 1)	Similar safety and immunogenicity profiles in both healthy and AD populations	[139]
IMVAMUNE	Human	Safe, well tolerated, and elicited strong immune responses; standard schedule elicited greater antibody titers	[140]
MVA vs. ACAM2000	Human (phase 3)	MVA induced higher neutralizing antibody titers and was associated with fewer severe adverse events.	[141]
CJ-50300	Human (phase 3)	95.0% exhibited cutaneous “take” reactions and 88.5% demonstrated humoral immunogenicity; 95.9% reported vaccine-related adverse events; no serious reactions.	[142]
ACAM2000	Human	Linked to a case of mother-to-child transmission	[146]
JYNNEOS	Animal	No evidence of fetal malformations or developmental delays	[148]

**Table 9 vaccines-13-00125-t009:** Vaccine platforms for mpox.

Therapy	Study Type	Outcomes	Reference
Tian Tan(first generation)	Human	Provided partial protection against mpox for over 42 years	[152]
MVA-BN/JYNNEOS(third generation)	Human	78% effectiveness in preventing symptomatic mpox 14 days after pre-exposure immunization	[154]
	Human	88.8% effectiveness when used for post-exposure prophylaxis	[155]
4pox DNA vaccine	Animal	Comparable protection to ACAM2000 and MVA	[157,158]
Multivalent mRNA vaccine	Animal	Induced T-cell responses and protected against vaccinia virus	[159]
Protein-based subunit vaccines	Animal	Demonstrated some efficacy in protecting against mpox	[160]

**Table 10 vaccines-13-00125-t010:** Studies assessing vaccines for HFMD.

Therapy	Study Type	Outcomes	Reference
Inactivated EV-A71 vaccine	Human (phase 3)	97.4% efficacy against HFMD	[163]
	Human (phase 3)	94.8% efficacy against HFMD; 100% effective in preventing hospitalization and neurological complications	[164]
	Human (phase 3)	Overall efficacy of 94.7% for two years	[165]
	Human (phase 4)	89.7% overall effectiveness over a 14-month follow-up	[166]
Bivalent (CVA6, CVA10)	Animal	Induced high levels of IgG and neutralizing antibodies; passive transfer of antisera from vaccinated mice to recipient mice potently protected recipient mice against CVA6 and CVA10 challenge	[168]
Trivalent (EV71, CVA6, CVA16)	Animal	Provided full protection from lethal challenge against EV71 and CVA16	[169]
Trivalent (CVA6, CVA10, CVA16)	Animal	Induced neutralizing antibody and cell-mediated immune responses	[170]
Tetravalent (EVA71, CVA6, CVA10, CVA16)	Animal	Elicited antigen-specific and long-lasting serum antibody responses; passively transferred vaccine-immunized sera conferred efficient protection against single or mixed infections	[171]

**Table 11 vaccines-13-00125-t011:** Studies assessing vaccines for GBS.

Therapy	Study Type	Outcomes	Reference
Trivalent (capsular polysaccharides Ia, Ib, III)	Human (phase 1b/2)	Safe; induced serotype-specific antibody production in women and higher antibody concentrations in infants	[180]
	Human (phase 2)	Safe; induced 8–16-fold increase in IgG concentrations	[181]
	Human (phase 2)	Serotype-specific IgG geometric mean concentrations were 13–23-fold higher in vaccine vs. placebo recipients on day 31 and persisted until postpartum day 90. Antibody transfer ratios were 0.62–0.82.	[182]
	Human	Immunogenic for all serotypes; well tolerated; antibody transfer ratios of 0.66–0.79	[183]
	Human (phase 2)	Elicited functional antibodies that were placentally transferred	[184]
	Human (phase 2)	A second dose administered 4–6 years after the first was immunogenic, with a favorable safety profile.	[185]
	Human (phase 2)	Lower antibody responses and reduced antibody transfer to infants in HIV-infected pregnant women	[186]
Hexavalent (Ia, Ib, II through V)	Human (phase 1/2)	Well tolerated; elicited robust immune responses	[187]
	Human (phase 2)	Safe; produced serotype-specific anti-CPS antibodies; antibody transfer ratios were 0.4–1.3. 57–97% of infants developed antibodies associated with a reduced risk of disease	[188]
Trivalent vs. hexavalent	Human	The trivalent and hexavalent conjugate vaccines have efficacies of 61.2% and 91.4%, respectively, for preventing STSS from GBS.	[176]
GBS-NN (AlphaC and Rib)	Human (phase 1)	Well tolerated; highly immunogenic. Antibody levels remained elevated for up to one year.	[189]
SL/pVAX1-sip	Animal	Safe and effective in tilapias; induced protective immune responses without genomic integration	[190]
GBS CPS III-TT conjugate vaccine	Human (phase 2)	Significantly delayed acquisition of vaginal and rectal GBS III colonization	[191]

## Data Availability

No new data were created or analyzed in this study.

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
