# Peer review of "Vaccines in Dermatology—Present and Future: A Review"

_vaccines, 2025, doi:10.3390/vaccines13020125_

Round 1

Reviewer 1 Report

Comments and Suggestions for Authors

The authors present an overview of novel aspects of vaccinations for skin conditions. The paper is interesting, well-written and covers a broad range of conditions.

I only have very minor comments:

1. The possibility for an acne vaccine is interesting. The authors mention trials being performed in individuals with acne. Is the vaccine meant to be used on persons with acne or as a prevention in any individual before developing acne. In the latter case how would selection be made?

2. Pox vaccines: Would you consider mentioning when the 1. generation, second and third generation vaccines became available. This would give an overview of the age groups having received them.

Author Response

Comment 1: "The possibility for an acne vaccine is interesting. The authors mention trials being performed in individuals with acne. Is the vaccine meant to be used on persons with acne or as a prevention in any individual before developing acne. In the latter case how would selection be made?"

Response 1: Thank you for pointing this out. We have added a line explaining that these vaccine candidates are meant for use in persons with acne, not as prevention. These changes can be found in page 4, lines 161-163.

Comment 2: "Pox vaccines: Would you consider mentioning when the 1. generation, second and third generation vaccines became available. This would give an overview of the age groups having received them."

Response 2: Thank you for the suggestion. We have accordingly included the years in which the vaccines discussed were introduced or approved for use: the third-generation vaccine MVA-BN/JYNNEOS in 2019 (page 16, line 699), first-generation vaccines in the 1950s (page 16, line 701), and the second-generation vaccine ACAM2000 in 2007 (page 17, line 717).

Reviewer 2 Report

Comments and Suggestions for Authors

Abstract must be improved in order to better explain the purpose and the methods of this paper. Furthermore, vaccines taken into consideration for this review should be listed in the abstract.

I suggest to add the most common side effects for these vaccines.

Author Response

Comment 1: "Abstract must be improved in order to better explain the purpose and the methods of this paper. Furthermore, vaccines taken into consideration for this review should be listed in the abstract."

Response 1: Thank you for your feedback. We have revised the abstract accordingly and listed the vaccines discussed in the abstract (page 1, lines 15-25)

Comment 2: "I suggest to add the most common side effects for these vaccines."

Response 2: Thank you for the suggestion. We have added the most common side effects reported.

  • Melanoma peptide-based vaccine: page 8, lines 370-371.
  • Warts: page 10, lines 467-469; lines 474-475; lines 483-484; lines 494-495
  • Candidiasis: page 12, lines 531-536
  • Leishmaniasis: LEISH-F3 (page 15, lines 648-650), phase 1 trial of ChAd63-KH (page 15, lines 659-660), phase 2 trial of ChAd63-KH (page 15, line 662)
  • Mpox: first-generation vaccines (page 17, lines 720-721),
  • GBS: trivalent vaccine (page 20, lines 827-829 and lines 834-835); hexavalent vaccine (page 20, lines 841-843 and 847-849), GBS III-TT vaccine (pages 20-21, lines 869-872)

Reviewer 3 Report

Comments and Suggestions for Authors

The authors undertook an ambitious goal - assessing the development and role of vaccines in dermatologic conditions. This reviewer appreciates the large amount of scholarship that went into this review. 

A minor comment is that the discussion of anogenital HPV vaccines was excellent, but nongenital HPV types are universal and cause morbidity. Consider discussing these.

Another is that smallpox was mentioned many times, especially in relation to Mpox. However, there was no separate discussion of this former nightmare. The elements of these vaccines were discussed, but not under its own subheading.

Author Response

Comment 1: "A minor comment is that the discussion of anogenital HPV vaccines was excellent, but nongenital HPV types are universal and cause morbidity. Consider discussing these."

Response 1: Thank you for your suggestion. We agree that discussing nongenital HPV types is valuable. In response, we have added a line noting that non-genital warts are extremely common (page 4, lines 184-185) and included two studies detailing the efficacy of the quadrivalent vaccine in treating recalcitrant acral warts (pages 5-6, lines 232-245).

Comment 2: "Another is that smallpox was mentioned many times, especially in relation to Mpox. However, there was no separate discussion of this former nightmare. The elements of these vaccines were discussed, but not under its own subheading."

Response 2: Thank you for your feedback. We agree that a separate discussion on smallpox is valuable and have added a dedicated section addressing this topic (pages 16-18).

Reviewer 4 Report

Comments and Suggestions for Authors

Your review brings into attention the used vaccines or that are in study for some dermatological diseases. It is a very interesting theme, who pointed out very well the vaccine’s mechanism of action, the indications and the side effects that may appear.   

Because some vaccines are still in research you need to insert that in the text and/or in the title (ex. Vaccines in Dermatology - present and future: a review).

The abstract it is not appropriate. Please redo.

Please remove the commercial name of some pharmaceutical company (ex. from the line 136, the VVZ vaccines, etc).

The conclusions are consistent with the review.

The references are appropriate.

Author Response

Comment 1: "Because some vaccines are still in research you need to insert that in the text and/or in the title (ex. Vaccines in Dermatology - present and future: a review)."

Response 1: Thank you for the suggestion. We have retitled the review accordingly and included a line in the abstract stating that some vaccines have been approved while others are currently being researched (page 1, lines 18-19)

Comment 2: "The abstract it is not appropriate. Please redo."

Response 2: Thank you for your feedback. We have revised the abstract (page 1, lines 15-25)

Comment 3: "Please remove the commercial name of some pharmaceutical company (ex. from the line 136, the VVZ vaccines, etc)."

Response 3: Thank you for the suggestion. We have removed the names of the pharmaceutical companies from line 136.

Round 2

Reviewer 4 Report

Comments and Suggestions for Authors

Dear authors,

You made the changes that I suggested.